# Impact of Gut Microbiota in Brain Ageing: Polyphenols as Beneficial Modulators

**DOI:** 10.3390/antiox12040812

**Published:** 2023-03-26

**Authors:** Fiorella Sarubbo, David Moranta, Silvia Tejada, Manuel Jiménez, Susana Esteban

**Affiliations:** 1Neurophysiology Lab, Biology Department, Science Faculty, University of the Balearic Islands (UIB), Crta. Valldemossa km 7.5, 07122 Palma, Spain; 2Research Unit, Son Llàtzer University Hospital (HUSLL), Crta. Manacor km 4, 07198 Palma, Spain; 3Group of Neurophysiology, Behavioral Studies and Biomarkers, Health Research Institute of the Balearic Islands (IdISBa), 07198 Palma, Spain; 4CIBERON (Physiopathology of Obesity and Nutrition), 28029 Madrid, Spain

**Keywords:** gut microbiota, gut–brain axis, brain, ageing, neurodegeneration, polyphenols, inflammation, nutrients, antioxidants, anti-ageing strategies

## Abstract

Brain ageing is a complex physiological process that includes several mechanisms. It is characterized by neuronal/glial dysfunction, alterations in brain vasculature and barriers, and the decline in brain repair systems. These disorders are triggered by an increase in oxidative stress and a proinflammatory state, without adequate antioxidant and anti-inflammatory systems, as it occurs in young life stages. This state is known as inflammaging. Gut microbiota and the gut–brain axis (GBA) have been associated with brain function, in a bidirectional communication that can cause loss or gain of the brain’s functionality. There are also intrinsic and extrinsic factors with the ability to modulate this connection. Among the extrinsic factors, the components of diet, principally natural components such as polyphenols, are the most reported. The beneficial effects of polyphenols in brain ageing have been described, mainly due to their antioxidants and anti-inflammatory properties, including the modulation of gut microbiota and the GBA. The aim of this review was, by following the canonical methodology for a state-of-the-art review, to compose the existing evidenced picture of the impact of the gut microbiota on ageing and their modulation by polyphenols as beneficial molecules against brain ageing.

## 1. Introduction

Due to the globally existing demographic context characterized by the increase in life expectancy and the large proportion of elderly people worldwide, one of the current great challenges in science is the search for strategies to prevent ageing [1]. In this context, within ageing, brain ageing is one of the most studied phenomenon, due to its medical, social, and economic impacts [2]. Anti-ageing strategies can be multiple, ranging from the most interventionists with pharmacological or genetic techniques, to the less interventionists that use the modification of lifestyle [3]. In this sense, diet is being increasingly studied because its benefits have been known since ancient times, because it is an easily modifiable element with a direct effect on general physiology [4,5].

Brain ageing is a complex physiological process that includes different mechanisms, some clearly described but others under study, which makes the final integration of all of them a complex scientific challenge. Among the main known causes of brain ageing and neurodegeneration are the limited renewal capacity of neural cells, neuronal/glial dysfunction [6], alterations of the brain vasculature and the blood brain barrier (BBB) [7,8], and loss of plasticity [9,10,11,12,13], accompanied by an age-dependent decline in the brain repair systems, including adult neurogenesis [14]. The causes of this damage are related to an increase in oxidative stress levels [15], together with a proinflammatory state [16,17], without the appropriate antioxidant and anti-inflammatory responses. In gerontology, this condition is defined as inflammaging [16]. The control of this state contributes to the prevention of ageing and to the repair of the brain after damage, playing a key role in complex brain functions, such as cognition (e.g., memory [18,19,20,21,22], learning [23]), mood [24,25], or sensorial ability (e.g., olfactory [26,27]).

In order to address brain ageing, there are different therapeutic strategies, one of which is to consider the modifications of the gut–brain axis (GBA) as therapeutic mediators [28]. During the 1960s and 1970s, several peptides were found in both the gastrointestinal tract and the brain; subsequently, the concept of amine precursor uptake and decarboxylation hypothesis (APUD) was developed. Over time, this evolved into the concept of the GBA, based on the predominant concept of a bidirectional communication between the gut and the brain [29,30], in health and in disease conditions [31,32]. In this bidirectional communication of the GBA, four pathways of communication from the brain to the gut are involved: the peripheral autonomic nervous system (sympathetic and parasympathetic) (e.g., enteric nervous system or the vagus nerve [33]), the neuroendocrine outputs axis (e.g., hypothalamic–pituitary–adrenal axis (HPA)), the neuroimmune systems (immune cells and glia [34]), and the systemic circulation [34]. There are four main groups of messengers participating in this communication, with the ability to modify the cerebral function and behaviour, these being microbial factors (e.g., short-chain fatty acids (SCFAs), branched chain amino acids and peptidoglycans [35]), gut hormones, cytokines, and sensory neurons (Figure 1). Among the microbial factors, the gut microbiota composition is notably, and this comprises several species of microorganisms, including bacteria, yeast, and viruses [36], that live in a delicate symbiosis. The disruption of this relation, known as dysbiosis and common in ageing, can lead to aberrant neural and glial reactivity, accompanied by the loss of brain functionality, observed at a cognitive level [37]. Thus, a functional relationship links the gut microbiota and GBA with brain functions [34]. Therefore, alterations in this axis not only affect the neural regulation of the gastrointestinal tract, but also several brain functions with the onset of neurological symptomatology, for instance in mood (e.g., depression, anxiety), in neurodevelopment (e.g., autism) [38,39], and in cognition (e.g., Alzheimer’s disease) [40,41,42]. It is important to note that imbalances affecting this complex ecosystem can impact the permeability of the body barriers, including the BBB and the enteric barrier [43].

Many factors can influence the microbiota composition and their functionality, both intrinsic to the organism itself and to external ones. Among the intrinsic factors, genetics and the ageing process significantly affect the composition of the microbiome [44,45]. External factors include infectious processes, drug administration (including antibiotics), environmental stressors, or factors related to lifestyle, especially diet. In this context, dietary polyphenols are pointed out as ductile molecules with the capacity to counteract the impact of ageing in the brain, due to their natural antioxidant and anti-inflammatory properties [46,47], both directly in the brain due to its ability to cross the BBB, and indirectly through modulation of the microbiome and GBA [48,49,50].

The aim of this review is to assess the known data related to the connection between the gut microbiota and GBA and brain ageing, with special attention on the impact that polyphenols may have on it. To reach this aim, a canonical methodology for a state-of-the-art review was followed [51]. A list of keywords (gut microbiota; gut–brain axis; brain; ageing; neurodegeneration; polyphenols; inflammation; nutrients; antioxidants; anti-ageing strategies) was initially identified. Then, different keyword combinations, each containing the term “ageing”, were used to search the following sources: PubMed, Embase, Medline, Scopus, Web of Knowledge, and Google Scholar. Articles published in English and indexed as original articles, meta-analysis reviews, narrative reviews, clinical cases, and comment to editor, with qualitative and quantitative data, were included in the analysis. Although the time range was not limited, the most recent publications were prioritized.

## 2. Evidence of the Impact of the Gut Microbiota in Brain Ageing

In the last few decades, gut microbiota have emerged as one of the key regulators of brain function, gaining attention for their role in brain health, brain ageing, and neurodegenerative disorders [52]. It has also been demonstrated that a crosstalk exits between gut microbiota and the ageing process [53]. A large portion of the experimental data that have demonstrated the impact of gut microbiota on brain ageing and neurodegenerative disorders are based on animal studies: germ-free (GF) animal data, infection or antibiotic drug use data, and microbiota transfer studies, but less observations from human studies can be found. Altogether, these studies support the hypothesis of a key role of the gut microbiota in the ageing process [54], and in central nervous system (CNS) disorders [55]. One of the main gut changes that has an impact on ageing is the alteration of the gut microbiota composition, accompanied by alteration of the enteric barrier. In this case, the presence of harmful microorganisms or substances could also be observed, accompanied by a loss of the beneficial ones in the microbioma environment. The main interactions between the changes in the gut microbiota and brain ageing are summarized in Figure 2.

(a) Alterations of the gut microbiota composition with an impact on brain ageing: Throughout life, many factors can influence the composition of the gut microbiota [56], such as the type of delivery at birth [57], genetics [58], infectious processes, antibiotic drug use, lifestyle, diet, stress factors, and the development of the ageing process [44,59]. It has been demonstrated that certain age-related changes in the composition of the gut microbiota are associated with many health conditions, including increased frailty, cognitive impairment, or depression, both in humans [44,60,61,62] and in animals [62,63]. During ageing, the gut can be dominated by noxious microorganisms, producers of metabolites with adverse effects, a phenomenon known as dysbiosis. These changes, related to the microbiome, have been described in many studies and have been attributed to an acceleration of the ageing process, in turn, generating impairment in different organs of the human body [54], including the brain [64]. An example of these alterations can be seen in the fact that some specific microbial taxa, such as the *Porphyromonadaceae* family, have been related to the appearance of cognitive and affective disorders [65,66,67], despite large individual variations. On the contrary, others microorganisms are associated with a reduction of frailty in older populations, such as the *Bacteroidetes* family [68,69,70], or the species *Clostridium cluster XIVa* and *Faecalibacterium prausnitzii* [68,70,71]. Furthermore, it has been proposed that age-associated characteristic inflammatory status can be, at least in part, promoted by age-associated dysbiosis [72,73]. As age advances, a greater proinflammatory state and a lower adaptive immune response are progressively expressed [74,75,76,77,78], which contributes to accelerating the aging process and increasing susceptibility to developing age-associated chronic diseases, some of which compromise brain health and promote CNS functional decline [79]. In this sense, the relationship between the alterations of the gut microbiota pattern and some brain alterations has been demonstrated, such as anxiety and depression [62], cognitive dysfunction [66], and the development of neurodegenerative diseases such as Alzheimer [80] and Parkinson [81]. Although inflammation may not normally represent a triggering factor in neurodegenerative diseases, the severity of the neuroinflammation is certainly a factor that accelerates the progression of cognitive decline and the development of neurodegenerative diseases [81,82,83].

(b) Alterations of the enteric barrier and BBB composition with an impact on brain ageing: The enteric/gut barrier and the BBB are two barriers that belong to the GBA, whose integrity is central for axis functionality. The enteric barrier is defined as a semipermeable tissue that allows the uptake of essential nutrients and the gut immune defence, while being restrictive against pathogenic molecules and bacteria from the intestinal tract. Both structural and molecular components of the barrier act together to fulfil this essential function of the gastrointestinal tract. Histologically, the apical layer from gut epithelium is the mucus layer, which is formed by a sieve-like structure overlying the intestinal epithelium. Antimicrobial peptides and secretory immunoglobulin A (IgA) molecules are secreted into the mucus layer as immune-detectors and regulatory proteins. Above this layer, the intestinal epithelial cells, which are tightly attached to each other by junctional complexes, make a continuous monolayer. The tight junctions are located at the apical side of the cells and regulate the transport of small molecules and ions. The adherens junctions and desmosomes provide strict cell-adhesion bonds and assist in the maintenance of the integrity of the intestinal barrier. The lamina propria contains immune cells from the adaptive and innate immune system (e.g., T cells, B cells, macrophages, and dendritic cells) that participate in the immunological defence of the intestinal barrier [84]. The gut barrier is the first defence against potential harmful agents that could have been ingested with food. This barrier constantly deals with innocuous and non-innocuous food antigens, and with harmful microorganisms. The gut barrier is equipped to interact with and/or tolerate the gut microbiota, induce systemic tolerance to food antigens, and fight against possible pathogens. Deficiencies in these functions can lead to intestinal disorders such as inflammatory bowel disease and irritable bowel syndrome, food allergy or intolerance, and microbial infection, which could favour a general body inflammatory state that also impacts brain functionality and accelerates the brain aging process [85,86].

In turn, the BBB is a term used to describe the exclusive properties of the microvasculature of the CNS, characterized for being a semipermeable and extremely selective vascular wall that separates blood from the brain’s extracellular fluid. This barrier is composed mainly of capillary endothelial cells (sealed by special tight junctions), astrocytes, and pericytes, as well as some other elements, such as basement membrane and other cell types that contribute to the immunological function. These components, which are frequently referred to as the neurovascular unit, preserve a healthy BBB to guarantee the appropriate CNS activity [87]. Although the enteric barrier and the BBB provide defence in very different environments, there are many similarities in their mechanisms of action. In both cases, there is a physical barrier formed by a cellular layer that tightly regulates the movement of ions, molecules, and cells between two tissue spaces. These barriers interact with different cell types, which dynamically regulate their function, and with different immune cells that assess the physical barrier and provide innate and adaptive immunity [88]. The general postulated hypothesis is that characteristic imbalances of ageing that affect the complex ecosystem of the gut microbiota contribute to the decrease in the normal function of these barriers, impacting the permeability in such a way that it allows the flow of potentially harmful substances into the brain tissue through the GBA [34,43,89], which in turn leads to chronic inflammation. In fact, during ageing, it has been demonstrated that the integrity and function of the gastrointestinal barrier weakens [43,72], negatively affecting BBB permeability and accelerating neuroinflammation and functional decline in the CNS [89].

(c) Metabolism-derived substances and immune cells with an impact on brain ageing: The metabolites generated in the gut as a result of digestion become circulating metabolites when they have passed the enteric barrier, and if they also cross the BBB they may enter into the CNS. These metabolites regulate the function of peripheral immune cells, which in turn may influence brain function directly or indirectly, for example via the cerebral lymphatic network [90]. This is the case in butyrate, which is produced through microbial fermentation of dietary fibres in the lower intestinal tract. Butyrate exerts its functions by acting as a histone deacetylase inhibitor or by signalling through several G protein-coupled receptors. This molecule has received particular attention for its beneficial effects on intestinal homeostasis and energy metabolism. Interestingly, butyrate enhances intestinal barrier function and mucosal immunity because of its anti-inflammatory properties [91]. Growing evidence has highlighted the impact of butyrate on the GBA, especially in the regulation of inflammation [92]. In this context, another important molecule derived from digestion is the amino acid tryptophan, which is metabolized by gut commensals, yielding compounds that affect innate immune cell functions. Tryptophan also acts on receptors that regulate the maintenance of the immune response, including lymphoid cells, promoting T helper 17 cell differentiation, and interleukin-22 production. In addition, some microbiota-derived tryptophan metabolites (e.g., indole-3-propionic acid and indole-3-aldehyde) have endothelial direct protective effects maintaining the vascular endothelium integrity and functionality, influencing the development of vascular inflammatory phenotypes [93]. Therefore, the immune system, formed by immune cells in the body and microglia cells in the brain, plays an essential role in maintaining tissue homeostasis, responding to these molecules, and also in infection and injury. In the brain, microglia are the main resident immune cell group; they constantly monitor the microenvironment and produce factors that influence surrounding neurons and astrocytes (another glial cell type with important functions for neuron activity). In brief, metabolites generated in the gut microbiota can modulate not only the peripheral immune system [56] and the CNS microglia activation [94,95], but are also important for the prevention of brain inflammation. Interestingly, both systems have been highlighted as being impaired in ageing [96]. This can be explained because, in the brain, microglial cells play multiple roles in the regulation of inflammatory responses and neuronal function, being pivotal in the pathogenesis of neurodegenerative diseases [97]. In line with this, the restoration of gut microbiota balance and inflammation can have beneficial effects on brain function and age-related diseases [98]. This idea was already pointed out by Metchnikoff more than a century ago [99]. Currently, several sequencing studies, along with those modifying the gut microbiota composition in animal models by means of the administration of antibiotics or probiotics or transferring the microbiota, not only supports the role of the microbiota in brain inflammation and diseases, but also offers new therapeutic perspectives aimed at a specific microbial modulation to attenuate ageing or brain pathologies, because the modifications of the gut microbiota have been reported to have protective effects not only on ageing [77], but also on learning, memory [100], and in the attenuation of neurodegenerative pathologies such as in Alzheimer’s disease [101].

## 3. Modulation of Brain Ageing by Polyphenols via the Gut Microbiota

The onset and development of the ageing process can be modulated by lifestyle factors, such as it the case of diet components that interact with the gut microbiota before reaching the brain [102]. The great interest related to this interaction is because diet is an easily extrinsic modifiable factor. Polyphenols are one of the leading anti-ageing diet components, natural compounds exclusively synthesized by plants with chemical features related to phenolic substances [103]. Polyphenols can be found in plant-derived food, and are promising anti-ageing molecules, especially for the brain, because they have the ability to cross the BBB; exert antioxidant and anti-inflammatory properties [104,105]; and generate positive effects on the preservation of monoaminergic neurotransmitters [106], cognitive and motor functions [107,108], and neurogenesis [109]. All these parameters and functions can alter the brain homeostasis, and can thus promote or prevent ageing [110]. When polyphenols are obtained from food, their arrival to the brain takes place by passing through the enteric barrier, so that all the interactions of these molecules in the gut have an impact on the GBA, in the brain functionality and finally in the development of the ageing process [111]. In turn, all the age-related changes in the gut, including pathological conditions and inflammation, among others [112], have an impact on the absorption, metabolism, and the arrival and effectiveness of the polyphenols in the brain [28]. The effects of polyphenols on gut microbiota have been shown in vitro, in vivo, and in human studies (for a review, see [113]), but the mechanisms by which polyphenols modulate the gut microbiota and have an impact on brain ageing are still unknown. The main demonstrated interactions between polyphenols, the gut microbiota, and GBA, with a beneficial impact on healthy brain ageing, are described in Figure 3.

(a) Direct effects of polyphenols on the gut microbiota: Gut dysbiosis is one of the most frequently found age-related gut disturbances [114]. In this condition, although the dietary intake would be rich in polyphenols, their absorption would be deficient. The concrete effect of polyphenols on the intestinal microbiota and the GBA is not yet know, but increased evidence suggests that these molecules exert selective effects on the gut microbial biodiversity, thus preventing dysbiosis [115]. Specifically, polyphenols would have a direct effect on the gut microbioma composition by affecting the bacterial growth and metabolism, favouring an increase in beneficial bacteria and inhibiting the proliferation of pathogenic bacteria [113,116,117]. This also occurs with other types of dietry components, for instance, the increases in carbohydrate consumption favours the presence of *Prevotella*, being the main bacteria in the gut microbial community, or protein and saturated fat consumption increases the presence of *Bacteroides* [118,119]. In the case of polyphenols, at least in humans, it has been demonstrated that there are some microorganisms that especially participate in the metabolism of polyphenols and, in turn, these molecules can enhance their abundance, such as *Flavonifractor plautii*, *Slackia equolifaciens*, *Slackia isoflavoniconvertens*, *Adlercreutzia equolifaciens*, *Eubacterium ramulus*, *Eggerthella lenta*, *Bifidobacterium* spp., or *Lactobacillus* spp. (for a complete list of bacteria, see [113]). Interestingly, some of them, such as *Bifidobacterium* spp. and *Lactobacillus* spp., contribute to the gut barrier protection and to the decrease in factors related to the gut and general body inflammation and oxidative stress, for example, by blocking with their metabolites the activation of the nuclear factor-kappa B (NF-κB) or reactive oxygen species (ROS), generating an antioxidative and anti-inflammatory status of the GBA, which would be an important factor for preventing ageing [120,121,122]. This is also the case of *Faecalibacterium prausnitzii*, which presents anti-inflammatory action by blocking NF-κB activation. *Lactobacillus* spp. showed a neuroprotective role of the gut microbiota, due to its probiotic and high antioxidant activity [123]. In addition, *Faecalibacterium prausnitzii* presents anti-inflammatory action by blocking NF-κB activation. *Lactobacillus* spp. showed a neuroprotective role of the gut microbiota, due to its probiotic and high antioxidant activity [118,124], while minimally affecting—or even increasing—the population of beneficial bacteria. Altogether, they prevent inflammatory processes, which is key in ageing. Interestingly, it has been demonstrated that the effects of polyphenols on bacteria are due in part to the differences in the bacterial wall composition, Gram-positive bacteria being more sensitive to polyphenols than Gram-negative bacteria [125].

(b) Indirect effects of polyphenols on the gut microbiota: Depending on the concentration and the chemical structure of the specific polyphenol, and whether it occurs in conjugated or free form, the digestion of polyphenols produces phenolic metabolites. These metabolites have an impact on the rest of the microbiota and on the GBA [117,126]. Using in vitro models, it has been described that polyphenol permeation through the BBB is dependent on the degree of lipophilicity of each polyphenol or its metabolites, the less polar (i.e., *O*-methylated derivatives) being capable of greater brain uptake than the more polar ones (i.e., sulphated and glucuronidated derivatives) [127]. The arrival of these metabolites to the brain promotes neuro resilience [128]. This occurs in several ways, for example, the use of resveratrol in humans, because the resveratrol-derived metabolites, named dihydroresveratrol and lunularin, present antibacterial activity on pathological bacteria, such as *Salmonella enterica*, *Enterococcus faecalis*, and *Escherichia coli* [129]. These metabolites are mainly absorbed in phase II of metabolism in the small intestine, although they can also reach the colon [130]. In other cases, polyphenols favour the growth of *Roseburia* sp., which produces, by fermentation of fibre butyrate in the colon [131,132], a SCFA with activity as histone deacetylase inhibitor, which has anti-inflammatory and memory positive effects in rodent models [133].

To a certain extent, the demonstrated worsening of cognition and motor coordination in ageing is due to the accumulation of molecules in the brain derived from oxidative stress (mainly ROS) and inflammatory status, which produces cytokines and interleukins (IL) [134]. This status, accompanied by a lack of sufficient physiological response to counteract it, is a consequence of neural wear caused over the years [135]. Therefore, in this context, one important indirect effect of polyphenols is their action neutralizing ROS by inhibiting the major ROS-forming enzymes [136], such as monoamine oxidase or xanthine oxidase [137,138]. Furthermore, polyphenols and their in vivo metabolites do not act as conventional hydrogen-donating antioxidants, but they may exert modulatory actions in cells through actions in the protein kinase and lipid kinase signalling pathways [139] and may even involve hormetic effects to protect neurons against the oxidative and inflammatory stressors [140]. A study evaluating 45 polyphenolic compounds indicated that whilst both the flavanols (+)-catechin and (−)-epicatechin failed to inhibit NADPH oxidase, their relevant methylated metabolites exhibited strong NADPH oxidase inhibition through an apocynin-like mechanism [141]. Interestingly, other apocynin-like phenolic compounds, such as ferulic acid, homovanillin alcohol, caffeic acid, tyrosol, and vanillic acid, were also observed to inhibit NADPH oxidase activity, therefore indicating that smaller polyphenols, more structurally related to some colonic metabolites, may also serve as novel therapeutic agents in neuroinflammation.

Furthermore, polyphenols also act as a chelate of metal ions (mainly iron and copper) involved in ROS reactions [142], thus regulating the redox metal homeostasis and preventing metal deposition and neurotoxicity, with important implications for age-related neurodegenerative diseases such as dementia, Alzheimer’s, or Parkinson’s disease [143].

Polyphenols and their in vivo metabolites activate cellular stress-response pathways, resulting in the upregulation of neuroprotective genes. For instance, the polyphenol quercetin has been reported to inhibit neuroinflammation by attenuating nitric oxide production and inducible nitric oxide synthase (iNOS) gene expression in microglia [144,145], preventing inflammatory cytokine production and neuronal injury [146,147]. Polyphenols can also activate the transcription factor cAMP-response-element-binding protein (CREB), which induces the expression of brain-derived neurotrophic factor (BDNF), a mediator of neurohormesis. Finally, polyphenols can also regulate the transcription factor NF-*κ*B, which can mediate adaptive cellular stress responses by reducing the expression of inflammatory cytokines. Activated SIRT1 may also inhibit NF-*κ*B and so can reduce the cellular stress response, altogether modulating genes that encode antioxidant enzymes and other stress-response proteins [148].

(c) Direct effect of the gut microbiota on the activity of polyphenols: The suitable homeostasis of the gut microbiota has an impact on the activity of polyphenol, because their bioavailability depends on it [149]. If polyphenols are well digested and their metabolites arrive at the brain, they can exert brain anti-ageing effects. Gut microbiota contribute to polyphenol xenobiotic metabolisms and bioactive metabolite production, because gut microbiota-derived metabolites from polyphenols have been shown to contribute towards the metabolism of dietary polyphenols, leading to the generation of de novo and potentially bioactive compounds [150]. More than 90% of dietary polyphenols are not absorbed in the small intestine and reach the large intestine; thus, gut microbiota is critically important in turning these polyphenols into bioavailable products [151]. In general, gut microbiota metabolizes glycosylated polyphenols into lower molecular weight phenolic compounds, such as small phenolic acids [130]. Indeed, these gut microbiota-derived polyphenolic metabolites are also essential bioavailable polyphenolic acids. Polyphenols have been shown to undergo various enzymatic processes by gut microbiota, through which the polyphenol derivatives are in a form capable of being absorbed or are even more bioactive [152,153]. Interestingly, it was also demonstrated that gut bacteria can metabolize polyphenols into neurotransmitters and bioactive metabolites with pro-survival and anti-inflammatory effects for the neurons [85]. Therefore, the protective effects of polyphenols also depend on how gut microbiota metabolize these compounds. In fact, based on in vitro studies and in vivo studies focusing on the effect on immunometabolism of microbiota-derived polyphenolic metabolites, specific metabolizing-bacteria have been described, depending on the type of polyphenol that needs to be metabolized, each metabolite having different immunomodulatory effects (for more information, see [150]).

Once they have arrived at the brain, polyphenols and their metabolites can attenuate oxidative and inflammatory damage, preserving cognitive function in the ageing brain [154] by suppressing the expression of harmful molecules and senescence-related genes [155]. In this sense, the reduction of concentrations of antioxidants in both serum and brain cells is inherent to ageing [156], leading to a decline in neural survival and age-related functionality worsening, justifying the need for antioxidant and anti-inflammatory supplementation [157]. As previously mentioned, polyphenols have the abilities of antioxidants [136] and chelation of metal ions involved in ROS reactions [142], and are regulators of redox metal homeostasis, preventing neurodegenerative diseases such as dementia, Alzheimer, and Parkinson [143]. It should be noticed that the effects are accompanied by the modulation of polyphenols of signalling pathways and factors involve in cell survival preservation and neurogenesis, including SIRT1, NF-κB, Nrf2, and Wnt/β-catenin [46]. Both SIRT1 and NF-κB [158,159] are involved in the modulation of the neutralization of oxidative stress and inflammation. In response to a proinflammatory stimulus (e.g., tumour necrosis factor-α (TNFα) or IL-1), via Toll-like receptor 2 (TLR2) or cytokine receptors, NF-κB is translocated to the nucleus and activates the transcription of a cascade of proinflammatory cytokines and chemokines to induce inflammatory responses [160]. Remarkably, the activation of NF-κB-regulated gene expression is modulated by post-transcriptional modifications, such as methylation, phosphorylation, or acetylation, which can be altered upon stimulation [160,161]. The acetylation of p65/RelA, a subunit of the NF-κB protein, is of particular interest because it can either potentiate or diminish NF-κB signalling, depending on the particular acetylated lysine residue [162]. Specifically, the acetylation of lysine 310 is critical for the full activation of NF-κB transcription potential, and this can be deacetylated by SIRT1, a deacetylase and pro-survival protein [163],that can prevent inflammation by the deacetylation of the NF-κB protein [164]. In hippocampus and during ageing, the acetylated form of NF-κB has been seen to be increased, suggesting a lack of the inhibitory effect of SIRT1 against the NF-κB signalling [165]. This contributes to an inflammatory response in the brain and ageing [166], which demonstrates a key role of SIRT1 as a mediator of cognitive decline in normal ageing [166]. In this sense, polyphenols because of their antioxidant and anti-inflammatory capacity, are described as good candidates for the modulation of SIRT1 and NF-kB, which would help in preventing brain functionality decline. This is the case in polyphenols, resveratrol, silymarin, quercetin, naringenin [165], chatechin, and a diet enriched with polyphenols, which in aged rats have been described to modulate SIRT1 and activate the NF-κB signalling pathway, with a positive impact in the recovery of working memory, episodic-like memory [167], and motor coordination [168], always in ageing. Resveratrol also recovers the monoamine levels in the hippocampus and striatum [165]. It has been described that polyphenols exhibit their beneficial properties through a set of mechanisms, including the potential to modulate the triggering of neuroinflammation associated with ageing, by reducing the acetylation of NF-κB, which in turn could be due to the rise in SIRT1 levels, in key regions for cognitive processes, such as the hippocampus, as has been explained previously. The increase in SIRT1 implies not only a decrease in neuroinflammation but also a relation to the regulation of the functionality of cognitive processes, because SIRT1 regulates the expression of the neurotrophins involved in the morphology and functionality of synapses, thus regulating synaptic plasticity, adult hippocampal neurogenesis, and cognition [165]. This is accompanied by the antioxidant effects of polyphenols in the brain, because they prevent the oxidation of enzymes, such as the tryptophan hydroxylase enzyme (TPH) and the tyrosine hydroxylase enzyme (TH), and the inhibition of monoamine oxidase enzyme A (MAO-A), which together favours the increase in the synthesis and accumulation of monoamines. Moreover, it has been demonstrated that resveratrol enhanced the cholinergic system and BDNF and CREB signalling pathways in the prefrontal cortex of an Alzheimer’s disease mouse model. This can also improve physical strength [169]. Although other effects described after chronic resveratrol treatments involving the contribution of the cognitive and motor improvement observed here cannot be proven, it can be postulated that the modulation of SIRT1 and the NF-κB transcription factor would be part of this improvement. Therefore, alterations in the gut microbiota with an impact on these molecular pathways are essential for the observation of the beneficial effects previously mentioned.

## 4. Future Research Lines

Despite all the effects shown, more basic, clinical, and translational research regarding the effect of polyphenos is needed, especially regarding its administration by oral intake [111]. The first basic factor to study would be the changes in the gut microbiota profile, depending on the polyphenol diet composition, including the type of components and the doses. Relating to the previous idea, another question would be how polyphenols interact with the brain, depending on the human gut microbiome biodiversity, because it varies among individuals depending on genetical and environmental factors, such as diet habits [40]. In addition, the exact degree and the way that polyphenols act to prevent dysbiosis and ageing should also be addressed. All these interactions affect the metabolome, and consequently significant differences in metabolite concentrations can be observed, even if subjects consume the same diet. Furthermore, it also influences the arrival to the brain of potential beneficial metabolites from polyphenols. This could be studied in the serum/plasma of subjects recruited in clinical trials, with the aim to prove the effect of polyphenols on ageing. Great efforts should be made in order to design clinical trials to test the pharmacokinetics, safety, and efficacy of polyphenol oral intake in relation to the prevention of brain ageing. In this sense, there are many clinical trials related to young or adult populations [170], but in the aging population they are still scarce. Some clinical trials in young and adult people have been performed in patients with Type 1 diabetes, in which cocoa flavanols improved cognition and hemodynamic responses [171]. In healthy adults (aged 40–60 years old, n = 101) with overweight and obesity, the long-term (24 weeks) supplementation with anthocyanin-rich *Aronia melanocarpa* extract (90 mg and 150 mg) improved psychomotor speed [172]. In a double-blind controlled trial in young subjects (n = 90) with sickle cell disease, spearmint extract intake (900 mg/d) improved working memory compared to the placebo group [173]. Regarding the few trials focused on older people, it was found that patients with mild cognitive impairment saw an improvement in age-related episodic memory impairment after the consumption of flavonoids (n = 215, age: 60–70 years old, 6 months, 258 mg/d) [174]. In a randomized controlled trial with older people (n = 37, 12-week), the consumption of cherry juice improved memory and performance during learning tasks [175]. Furthermore, cognitive performance and attention were improved (n = 100, 12-week, 800 mg/d) after *Lactobacillus plantarum* (C29-fermented soybean) consumption [176]. The same effect was observed when *Cosmos caudaus* supplement (n = 23, 500 mg/d) was used [177]. Finally, another study showed that *Persicaria minor* extract supplement rich in polyphenols (n = 36, 6-month, 500 mg/d) improved visual memory, negative mood, and bilateral dorsolateral prefrontal cortex activation in this type of patient. However, there is a lack of trials focusing on non-pathological ageing situations. Therefore, more clinical trials are needed, such as the MaPLE trial [178], which found changes in serum metabolome in healthy elderly people after the consumption of a diet enriched with polyhpenols over 8 weeks.

On the other hand, animal models could also be interesting to study the amount of metabolites of polyphenols in the brain after oral consumption of polyphenols, or how this affects monoamine concentration as they are brain activity modulators. They could also be used to investigate the molecular pathways involved in ageing, including the expression of the main proteins involved in inflammaging. Moreover, the study of the effects of polyphenols on metagenomics, anatomy gut diversity, and the relation with metabolomics should also be addressed. For example, several studies have indicated a high interindividual variability, at least in humans, regarding polyphenol metabolism, two profiles of people being recognized regarding their response to metabolizing polyphenols in the gut: “producers” and the “non-producers”. The first group would be people who produce metabolites (e.g., equol and O-desmethylangolensin from isoflavones), and the “non-producers” would be people who do not produce them [179,180]. Finally, it could be addressed if the described effects of polyphenols at the molecular level in brain cells are due to an indirect mechanism of action on SIRT1, or to an enzyme direct action, as occured in the in vitro experiments [181,182]. Thus, the knowledge of the mechanism of action of polyphenols would help in the design of new drugs and more specific and effective anti-ageing therapies.

## 5. Conclusions

The gut microbiota and the brain are bidirectionally communicated via the GBA. This implies that changes in this communication are the causes of the loss or the gain of brain homeostasis and, consequently, they have an impact on brain ageing. There are extrinsic factors, such as diet, which have the ability to affect this gut–brain connection. Natural components of diet, such as polyphenols, due to their antioxidants and anti-inflammatory properties, have been highlighted as modulators of brain ageing, one of the ways being the regulation of gut microbiota and the GBA. Numerous studies in animals and some in humans have shown these favourable effects of polyphenols on brain ageing via the gut microbiome and GBA, indicating the need for further research in order to develop therapeutical strategies against ageing based on the oral intake of polyphenols. This topic is an actual and prevailing research line, focusing on the prevention of brain ageing.

## Figures and Tables

**Figure 1 antioxidants-12-00812-f001:**
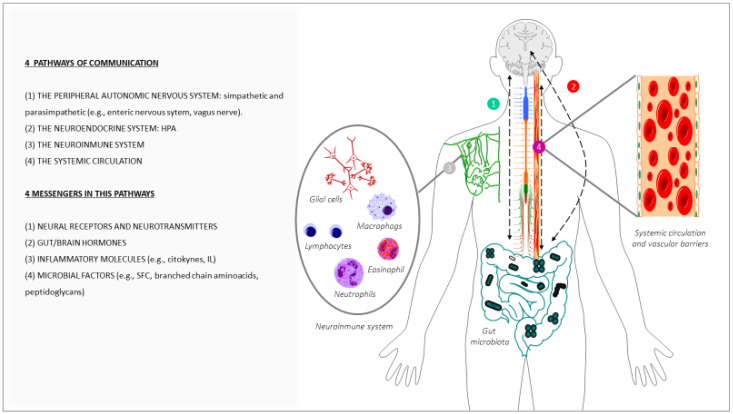
Main messengers and pathways of communication involved in the crosstalk between the gut and the brain.

**Figure 2 antioxidants-12-00812-f002:**
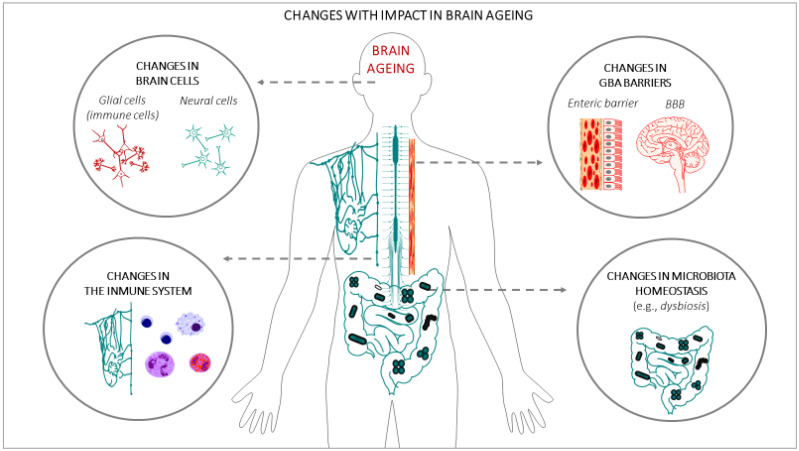
Changes in body systems with an impact on brain ageing.

**Figure 3 antioxidants-12-00812-f003:**
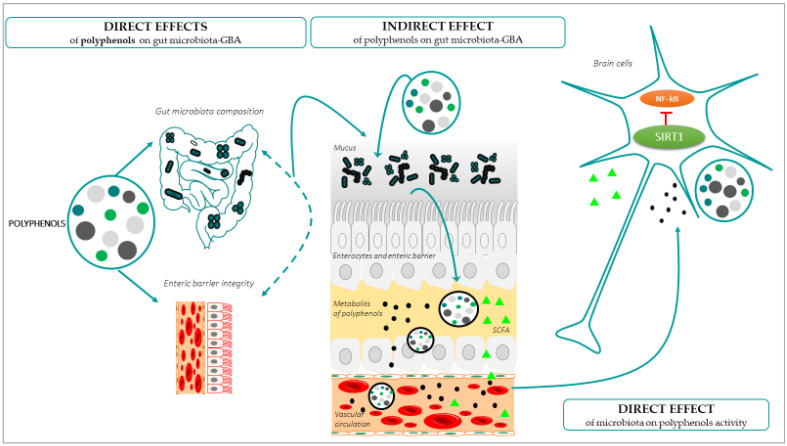
Direct and indirect effects of polyphenols on gut microbiome and how these changes directly affect brain aging.

## Data Availability

Not applicable.

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
