# Peer review of "Impact of Gut Microbiota in Brain Ageing: Polyphenols as Beneficial Modulators"

_antioxidants, 2023, doi:10.3390/antiox12040812_

Round 1

Reviewer 1 Report

The review provides an overview on the role of the gut microbiota in brain aging processes and the protective role of polyphenols. This aspect of the gut-brain axis is a current and relevant topic in microbiome research. Overall, the review is structured, interesting to read, and relatively well-written. The figures are illustrative and explanatory. However, I have some concerns that should be addressed by the authors when revising their manuscript.

1. Terms like “second brain” should be avoided since they are ill defined.

2. The review lacks a detailed description of the gut-vascular barrier and the blood-brain barrier with a clear focus on what is known on the microbiota-dependent regulation of these anatomical sites. This is an important part of the presented concept.

3. There is a relevant review on microbial metabolites and polyphenols to which the authors may refer to in the description of section 3 (PMID: 33036205).

4. The review would benefit from a table listing the involvement of specific gut microbes in metabolizing polyphenols.

5. The authors should also mention the influence of tryptophan metabolites on the vascular system and the capacity of these metabolites of crossing the blood-brain barrier (PMID: 35451695).

6. In contrast to the role of polyphenols in the prevention of aging, the role of the microbiome in these processes is not well-elaborated. The link of microbiome to mechanisms involved in aging needs to be better worked out.

Author Response

Reviewer #1

Point 1: Terms like “second brain” should be avoided since they are ill defined.

Our response: According to the Reviewer suggestion we have, in section 1. Introduction, deleted the term “second brain”, which was mentioned one time in this way: “Consequently, the GBA has been portrayed as the "second brain". We agree with the Reviewer that although it is a didactic and frequently used term, which explains well the interaction between brain and gut, perhaps it should be avoid in the framework of a scientific review. Therefore, this term was deleted.

Point 2: The review lacks a detailed description of the gut-vascular barrier and the blood-brain barrier with a clear focus on what is known on the microbiota-dependent regulation of these anatomical sites. This is an important part of the presented concept.

Our response: In accordance with the reviewer’s recommendation in section 2. Evidence of the impact of the gut microbiota in brain ageing, it was introduced a new paragraph b) Alterations of the enteric barrier and BBB composition with impact in brain ageing. In this part new information about the enteric barrier and the blood brain barrier were added, considering it a fundamental part in the defense against pathogens and as the main barriers for the passage of substances, before reaching the brain. All essential in the regulation of the gut brain axis and ageing. Therefore, there were added the definitions of both concepts, including the structure, and information about the microbiota-dependent regulation of barrier. See Section 2, from line 122 to line 156.

Point 3: There is a relevant review on microbial metabolites and polyphenols to which the authors may refer to in the description of section 3 (PMID: 33036205).

Our response: Following the Reviewer’s suggestion in section 3. Modulation of ageing by polyphenols via the gut microbiota, paragraph c) Direct effect of microbiota on polyphenols activity, we have added information cited in the manuscript[1], explaining how gut microbiota influences the metabolism of polyphenols and also their role in immunometabolism associated with the prevention of inflammation and ageing. See from line 273 to line 289.

Point 4: The review would benefit from a table listing the involvement of specific gut microbes in metabolizing polyphenols.

Our response: Thanks to the Reviewer’s comment number 3, we have found the following cite[1], and observing it, we have seen a table containing information about the metabolizing bacteria of the main polyphenols. Therefore, considering their importance but also taking into account that it is not the purpose of this article to make a detailed study of this aspect, it was decided to include the reference of the previous mentioned work in the text. In such a way, that readers could found the table if they are interested in this topic. Avoiding also duplication of information. See from line 286 to line 290.

Point 5: The authors should also mention the influence of tryptophan metabolites on the vascular system and the capacity of these metabolites of crossing the blood-brain barrier (PMID: 35451695).

Our response: Following the Reviewer’s suggestion in section 2. Evidence of the impact of the gut microbiota in brain, it was added the reference[2] in a new section named c) Metabolism-derived substances and immune cells with impact in brain ageing. In this part a brief summary was added about the effect of the microbiota-derived tryptophan metabolites in the regulation of immunity, for example modulating immune cells, which in turn modulates the triggering of inflammation, then affecting brain ageing. See Section 2, from line 166 to line 172.

Point 6: In contrast to the role of polyphenols in the prevention of aging, the role of the microbiome in these processes is not well-elaborated. The link of microbiome to mechanisms involved in aging needs to be better worked out.

Our response: first of all we appreciate the reviewer's comment since it helps us to improve the section 2. Evidence of the impact of the gut microbiota in brain ageing. For clarity, we have separated in sections the demonstrated main interactions between the changes in the gut microbiota, with impact in brain ageing, being the following ones: a) Alterations of the gut microbiota composition with impact in brain ageing; b) Alterations of the enteric barrier and BBB composition with impact in brain ageing; c) Metabolism-derived substances and immune cells with impact in brain ageing. Each section specifies the information found about that topic. This change also provided structural coherence with the section 3 of the article, since also in that section the information is separated in several paragraph following the same reasoning that in the section 2. Besides, this change has allowed us to include more information about the GBA barriers, also mentioned by the reviewer in the comment number 2, and to include information about tryptophan metabolites, proposed by the reviewer in the comment number 5. There was also added in the section c) information about other important metabolites, such as butyrate with impact in ageing and inflammation, and about the relationship of immune cells (general immune system and microglia cells) in the context of the impact of gut microbiome changes and brain ageing. In this point, and also for consistency with the text it was also improved the figure 2, citing all the main elements discussed in this section (Changes in the gut microbiota, in GBA barriers, in immunity). Therefore, taking all the changes in consideration the link of microbiome to mechanisms involved in brain aging was better worked out.

Reviewer 2 Report

The manuscript by Sarubbo et al. is a review of the impact of polyphenols on the but microbiota with respect to aging. Although the subject is an important one, this topic has recently been reviewed by several others and so the novelty may not rise to the level of a high impact journal like Antioxidants.

Major problems:

With that said, the manuscript may be suitable for submission to another journal but would require a major revision. Generally speaking, the English writing is so poor that the manuscript is almost unreadable and to some extent, it looks like it was written in the author's native language and then run through google translate. Moreover, there are paragraphs that are a page and a half long and sentences that are almost a paragraph long so its not just a matter of proper English translation, its also a general writing issue. The manuscript needs major editing by a native English speaker or at least somebody that is very fluent in English and a proficient writer before it could be reconsidered by another MDPI journal.

Another major issue is that most of the references are other reviews. This can be done sparingly, when addressing a general concept but original work should be the primary source of a review.

Minor problems:

References are incorrectly formatted. For example, on line 47, each references is listed individually. They should be listed as [10-14]. Some references are list in a random order, but should be listed in increasing order and finally some references are listed multiple times for the same sentence (Line 311, "In hippocampus and during ageing, the acetylated form of NF-κB has been described to be increased [156][156][156] suggesting a lack of the inhibitory effect of SIRT1 against the NF-κB signaling[156][156][156]. "

There are several technical writing problems. For instance, abbreviations are used incorrectly in many instances. For example, on line 300, (TLR-2)Toll-like receptors should be written as Toll-like receptor (TLR)-2.

If there exists contrary or conflicting research, this needs to be addressed.

Author Response

Reviewer #2

Point 1: With that said, the manuscript may be suitable for submission to another journal but would require a major revision. Generally speaking, the English writing is so poor that the manuscript is almost unreadable and to some extent, it looks like it was written in the author's native language and then run through google translate. Moreover, there are paragraphs that are a page and a half long and sentences that are almost a paragraph long so it’s not just a matter of proper English translation, it’s also a general writing issue. The manuscript needs major editing by a native English speaker or at least somebody that is very fluent in English and a proficient writer before it could be reconsidered by another MDPI journal.

Our response: After the three reviewer’s comments and suggestions the manuscript has been editorially revised, and long sentences or paragraphs were avoided. New paragraphs were added in section number 2. Evidence of the impact of the gut microbiota in brain ageing.  Also new references and information was added throughout the entire manuscript, gaining in clarity and coherence. Besides, a native English speaker has revised the last version of the manuscript. For changes see the text highlighted in green. Regarding the English correction, it is not highlight because all the manuscript was revised, and in order not to confuse with the rest of the changes it was decided not to add another color. A section of Acknowledgments was added, in order to thanks the English native speaker that revised the English language.

Point 2: Another major issue is that most of the references are other reviews. This can be done sparingly, when addressing a general concept but original work should be the primary source of a review.

Our response: Throughout the manuscript, references were corrected, avoiding duplication of information and adding other references related with original research articles.

Point 3: References are incorrectly formatted. For example, on line 47, each references is listed individually. They should be listed as [10-14]. Some references are list in a random order, but should be listed in increasing order and finally some references are listed multiple times for the same sentence (Line 311, "In hippocampus and during ageing, the acetylated form of NF-κB has been described to be increased [156][156][156] suggesting a lack of the inhibitory effect of SIRT1 against the NF-κB signaling[156][156][156]. "

Our response: The style of the references was corrected, avoiding duplications, and changing the mistakes mentioned by the reviewer. According to the Guide for Authors of Antioxidants, it was used the APC style.

Point 4: There are several technical writing problems. For instance, abbreviations are used incorrectly in many instances. For example, on line 300, (TLR-2)Toll-like receptors should be written as Toll-like receptor (TLR)-2. If there exists contrary or conflicting research, this needs to be addressed.

Our response: In accordance with the Reviewer’s quotation, we have corrected in section 3. Modulation of ageing by polyphenols via the gut microbiota, the following sentence “(TLR-2)Toll-like receptors” by “Toll-like receptor (TLR2)”. We have corroborated in other published[3] works that it is usually cite in this way: TLR2, without the hyphen between letters and number.  Besides, we have checked all the manuscript searching this type of mistakes and there were corrected throughout the entire text. For example, it was also changed in section 1. Introduction the following sentence:  “APUD (the amine precursor uptake and decarboxylation) hypothesis” by “the amine precursor uptake and decarboxylation hypothesis (APUD)”. Finally,  all the abbreviations are alphabetically list at the end of the manuscript.

Reviewer 3 Report

Manuscript No Antioxidants-2272268

Impact of The Gut Microbiota in Brain Ageing: Polyphenols as Beneficial Modulators” for Antioxidants

Comments:

1.      Paragraph 2. Please indicate in more detail what elements (cellular and soluble mediators) are involved in interactions with the gut microbiota and affect CNS activation or maintaining homeostasis.

2.      Paragraph 3. Polyphenols are quite a wide group of substances. Please add information whether any of the fractions belonging to polyphenols are particularly involved in the described processes.

3.      Readers may be interested in knowing whether and which polyphenols can reverse the adverse changes associated with brain aging. Please give a very brief mention about it.

Author Response

Reviewer #3

Point 1: Paragraph 2. Please indicate in more detail what elements (cellular and soluble mediators) are involved in interactions with the gut microbiota and affect CNS activation or maintaining homeostasis.

Our response: According to the request we added information about the elements that are involved in the interaction between the gut microbiota and CNS, metabolites like butyrate or tryptophan, and cells as the immune cells (general immune system and microglial cells) in section 2. Evidence of the impact of the gut microbiota in brain ageing, paragraph c) Metabolism-derived substances and immune cells with impact in brain ageing. See from line 157 to line 188.

We would like to comment that another Reviewer has suggested to improve section 2, therefore we have separated in this section, in paragraphs the demonstrated main interactions between the changes in the gut microbiota, with impact in brain ageing, being the following ones: a) Alterations of the gut microbiota composition with impact in brain ageing; b) Alterations of the enteric barrier and BBB composition with impact in brain ageing; c) Metabolism-derived substances and immune cells with impact in brain ageing. Each section specifies the information found about that topic. See from line 100 to 188. This change also provided structural coherence with section 3 of the article, since also in this section the information in separated in several paragraph following the same reasoning that in the section 2. There was also added in the section c) information about other important metabolites such as butyrate with impact in ageing and inflammation, and about the relationship of immune cells (general immune system and microglia cells) in the context of the impact of gut microbiome changes and brain ageing.

Point 2: Paragraph 3. Polyphenols are quite a wide group of substances. Please add information whether any of the fractions belonging to polyphenols are particularly involved in the described processes.

Our response: As Reviewer suggested we have added in section 3. Modulation of ageing by polyphenols via the gut microbiota, paragraph b) Indirect effect of polyphenols on gut microbiota, information about the metabolites of polyphenols produced by bacteria in the gut, and how they act in relation with the prevention of aging, principally by exerting in the brain antioxidant and anti-inflammatory effects. See from line 234 to line 272.

Point 3: Readers may be interested in knowing whether and which polyphenols can reverse the adverse changes associated with brain aging. Please give a very brief mention about it.

Our response: Regarding this comment, we take advantage of the new information added due to the comment number 2, to specify which polyphenols and in which manner they can prevent ageing. See all section number 3. Modulation of ageing by polyphenols via the gut microbiota. And particularly paragraph c) Direct effect of the gut microbiota on the activity of polyphenols, for example there is mentioned the effect of resveratrol. Silymarin, quercetin, naringenin, chatechin and a diet enriched with polyphenols. See line 273 to 290.

Round 2

Reviewer 1 Report

The authors have addressed my comments and the manuscript can be accepted for publication. However, a thorough final check of the language is needed and the respective titles of the referenced papers need to be added.